# Microstructure and Properties of Ti-Zr-Mo Alloys Fabricated by Laser Directed Energy Deposition

**DOI:** 10.3390/ma16031054

**Published:** 2023-01-25

**Authors:** Jingtao Zhang, Cunshan Wang, Nisha Shareef

**Affiliations:** Key Laboratory of Materials Modification by Laser, Ion, and Electron Beams, Dalian University of Technology, Dalian 116024, China

**Keywords:** titanium alloys, Ti-Zr-Mo composition design, laser directed energy deposition, microstructure, properties

## Abstract

The binary Ti-Zr congruent alloys have been a potential candidate for laser-directed energy deposition owing to an excellent combination of high structural stability and good formability. To solve its insufficient strength, a new series of Ti-Zr-Mo alloys with different Mo contents were designed based on a cluster model and then made by laser-directed energy deposition on a high-purity titanium substrate. The effect of Mo content on the microstructure and properties of the L-DEDed alloys was investigated. The consequences exhibit that the microstructure of all designed alloys is featured with near-equiaxed β grains without obvious texture. However, increasing Mo content induces a gradual refinement of the grain and a steady decrease in the lattice constant, which effectively improves the hardness, strength, wear and corrosion resistance of the designed alloys, but slightly weakens ductility and formability. From the viewpoint of both properties and forming quality, the Ti_60.94_Zr_36.72_Mo_2.34_ (at.%) alloy owns a proper match in mechanical, tribological, chemical, and forming properties, which is widely used in aeroengine components.

## 1. Introduction

Titanium alloys have been one of the most important materials in the biomedical, aerospace, and defense industries, because of their superior biocompatibility, excellent corrosion resistance, and specific strength [1,2,3]. Until now, titanium alloy components have been mainly fabricated through traditional manufacturing processes such as casting or forging. However, these techniques face great challenges in fabricating complex components, which results from complex and unusual deformation of titanium alloys, coupled with low thermal conductivity and low volume-specific heat [4,5,6]. Laser-directed energy deposition (L-DED), an additive manufacturing (AM) technology, provides a revolutionary approach for manufacturing the complex structure of titanium alloy since it has high fabrication freedom that allows for the building of various geometric components without additional equipment [7,8]. Therefore, many investigations have been conducted to investigate the L-DEDed titanium alloys, most of which focus primarily on traditional titanium alloys, especially TC4 alloy [9,10,11]. These alloys were originally designed for casting or forging applications, without involving the extraordinary metallurgical features of the L-DED process. The ultrafast cooling rate and steep temperature gradient of L-DED induce exclusive coarse columnar grains, leading to undesirable anisotropy properties [12,13,14,15]. Moreover, the multiple thermal cycles easily promote the local precipitation or decomposition of the metastable phase, resulting in an inhomogeneous microstructure and properties [9,16,17,18]. The adjustment of processing parameters shows a limited effect on these microstructure defects, limiting the potential advantage of L-DED in fabricating high-performance titanium alloys. Therefore, it is imperative to develop new titanium alloys suitable for L-DED.

The stability and morphology of microstructure, together with formability, should be considered for the alloy design of L-DED [19]. Structure stability strongly depends on the compatibility between the melt and solid structure, which represent whether the chemical short-range ordered structure can remain stable from melt to solid. The heritability and stability of microstructure largely depend on this. The thermal undercooling and constitutional supercooling of the local molten pool are two important factors affecting the microstructure morphology. During L-DED, the steep temperature gradient will induce the formation of columnar crystals making the material anisotropic, which could be relieved by enough constitutional supercooling. Owing to the non-homogenous nucleation, constitutional supercooling promotes the formation of equiaxed crystals. Good formability is associated with low metallurgical defect susceptibility. The binary Ti-Zr congruent alloy is a potential candidate for L-DED alloy since it solidifies at a constant temperature like pure metal [20], which gives it an excellent combination of high structural stability and good formability [21,22,23]. Unfortunately, the strength of the alloy was insufficient, resulting from the limited solid solution strengthening of a single alloying element. However, multi-element alloying is an effective way to solve a series of problems in congruent alloys [24].

Generally, the Mo element is a stronger stabilizer and a powerful solution-hardening agent of the β solid solution. One can expect that Mo addition will further increase the stability of the β phase and effectively enhance the strength of the Ti-Zr congruent alloy [25]. What is more important is that Mo can increase constitutional supercooling in front of the solidification interface by improving the growth restriction factor of the alloy, which will be beneficial to the formation of equiaxed grains and inhibit the epitaxial growth of crystals. Therefore, the newly designed Ti-Zr-Mo alloys with varying Mo addition were designed by the advanced cluster model and then deposited through L-DED technology on a high pure titanium substrate. The microstructure and property evolutions of the L-DED Ti-Zr-Mo alloys with Mo content were investigated. The goal is to optimize the structure and properties of the Ti-Zr congruent alloys by adjusting the addition of the Mo element, so as to provide a material basis for L-DED titanium alloys.

## 2. Design of Ti-Zr-Mo Alloys

It is well known that the special composition of an alloy results from its chemical short-range order structure [26]. Therefore, a cluster-plus-glue-atom model describing the structural characteristics of the short-range order is proposed. According to the model, the short-range order of any alloy can be simplified as the cluster part which is the first nearest neighbor polyhedron and the glue part located in the second nearest neighbor. In this model, the cluster atoms have a strong interaction with the solvent, while the glue atoms fill the gaps between isolated clusters and have a weak interaction with the solvent. Therefore, the model could be expressed in [cluster] [glue atoms] × [27]. It should be noted that the cluster formula is very similar to the molecular formula. However, it is not a formula in the traditional sense, because the cluster structural units are still interconnected by inter-atomic forces, which constitute the atomic and electronic structural units of an alloy with complex composition. Therefore, the model can be used for the composition design of multi-component alloys [28].

For the Ti-Zr congruent alloy, there are two CN14 type clusters in its local structure: a Zr-centered CN14 [Zr-Ti_14_] cluster and Ti-centered CN14 [Ti-Zr_14_], which is derived from the β-Ti and the β-Zr solid solutions, respectively, as shown in Figure 1. Based on theoretical calculation, the ideal cluster structural units of the two solid solutions are [Zr-Ti_14_]Ti_1_ and [Ti-Zr_14_]Zr_1_, and the alloy has the most stable structure when the atomic ratio of the two units is 10 to 6. Thus, the cluster formula of the Ti-Zr congruent alloy with the most stable structure can be expressed as 10[Zr-Ti_14_] Ti_1_ + 6[Ti-Zr_14_]Zr_1_ = Ti_156_Zr_100_ (here, these numbers represent the percentage of atoms).

After the Mo element was added to the Ti-Zr congruent alloy, it will enter the cluster structural unit of [Zr-Ti_14_]Ti_1_ and replace Zr to take up the central position of the cluster according to similar atom substitution and the cluster close-packing principles. In this way, the 10[(Zr_1-x_Mo_x_)-Ti_14_]Ti_1_ + 6[Ti-Zr_14_]Zr_1_ could be used to describe a new cluster formula of all designed alloys. Based on the cluster formula, five Ti-Zr-Mo alloys were designed, and their detailed composition is shown in Table 1.

## 3. Experimental Procedures

According to the composition of Table 1, the powders Ti, Zr, and Mo elements having 99.5~99.9% mass purity and 50~120 μm granularity were blended in a vacuum ball grinder at a rotational velocity of 120 rpm for 12 h, which were used as L-DED feedstock. To obtain a good interface between deposited alloy and substrate and reduce the influence of substrate dilution on the composition of deposited alloys, a high-purity titanium plate with a mass purity of 99.90% and a size of 70 mm × 50 mm × 10 mm was considered as substrate material.

The deposition process was performed using an LDM-8060 additive manufacturing system (RAYCHAM, Nanjing, China), which is mainly outfitted with a 6 KW fiber laser, a PC numerical manage multi-axis movement system and an automated powder feeder as shown in Figure 2. In the process of deposition, argon was used as a shielding gas and the oxygen content in the chamber was less than 100 ppm. Based on previous technological research, the optimal processing parameters were as follows: laser power 2 kW, laser beam diameter 4 mm, laser scanning speed 10 mm/s, power feed rate 0.8 r/min, overlapping rate 50%, and Z-axis growth 0.5 mm. Under these parameters, as-deposited alloys were fabricated with dimensions of 50 mm × 30 mm × 10 mm by bidirectional scanning.

The as-deposited alloys were sectioned into pieces along the built direction and then ground with sandpaper and polished to analyze the microstructure. The constituent phases of the designed alloys were identified using Empyrean X-ray diffraction (XRD) (PANalytical B.V.) equipped with Cu-Ka radiation with a scanning rate of 4 (deg.)/min over a range of 2θ angles from 20 to 100 (deg.). Microstructure evolutions were observed by a SU5000 scanning electron microscopy (SEM) (HITACHI, Tokyo, Japan) and the composition content was detected with an EPMA-1720 Electron Probe Microanalyzer (EPMA) (JEOL, Tokyo, Japan). To investigate crystal growth orientation, electron backscattered Diffraction (EBSD) (Thermo Fisher Scientific, Waltham, MA, USA) analyses were performed using a Hikari camera. Microhardness was tested by an HV-1000 micro-hardness tester (TIMESUN, Beijing, China) at a load of 10 N for 15 s. To obtain reliable data, 46 different points were tested for each as-deposited alloy, and their arithmetic mean value was taken. To test the compressive properties at room temperature, the samples from the central of L-DED alloys were machined into a cylinder 6 mm in length (building direction) and 3 mm in diameter and then were performed in UTM5504-G testing equipment (HST, Jinan, China) with a tensile rate of 0.1 mm/min. After the compressive deformation, the fracture morphologies were observed by SEM to investigate the fracture mechanism of the designed alloys. A CETR UMT-2 reciprocating friction-wear testing machine (CETR, USA) was used to measure the tribological properties of the as-deposited alloys. The studies were carried out with an oscillating stroke of 5 mm, a constant load of 10 N, a sliding velocity of 0.5 m/s, and a worn time of 0.5 h. A Si_3_N_4_ ball with 4.75 mm thickness and HV1500 in hardness was selected as the wear couple. The electrochemical properties were tested by an electrochemical system named M352. The work was equipped with a three-electrode battery, with the designed Ti-Zr-Mo alloy as the working electrode, a platinum sheet as the opposite electrode, and a saturated calomel electrode (SCE) as the reference electrode. With the corrosive medium of 1 mol/L HCl solution, a constant test temperature of 20 ± 1 °C and a voltage range of −0.5 V to 1.5 V at a scanning speed of 1 mV/s, dynamic potential polarization tests were carried out. The passive films were analyzed using a multi-technique X-ray photoelectron spectrometer (XPS, K-Alpha+) (Thermo Fisher Scientific, Waltham, MA, USA) with an optimal spatial resolution of ≤30 μm. Optical microscopy (OM, OLYMPUS) (LEICA DMi8, Weztlar, Germany) was used to assess the density of as-deposited Ti-Zr-Mo alloys. Four cross-sections were cut parallel to the direction of deposition and treated by metallographic preparation procedures for observation. The oxygen content in the raw powders was tested using Axios X-ray fluorescence spectrometry. A 3D Surface Optical Profiler (Zygo 9000) (ZYGO Corporation, Middlefield, CT, USA) was used to measure the surface roughness (Ra) of the designed alloys.

## 4. Results and Discussion

### 4.1. Microstructure

Figure 3 presents XRD patterns of the designed alloys. For comparison, the pattern of the Ti-Zr congruent alloy is also shown in the figure. The conclusion is that all designed alloys consist only of β solid solution, which differs from the Ti-Zr congruent alloy having a dual-phase structure of α and β. This suggests that Mo is an effective stabilizer of the β solid solution. Such stable characteristics can be evaluated through Mo equivalence (Mo_eq_). According to the Mo equivalence formula, Mo_eq_ = Mo + Nb/3.3 + Ta/4 + W/2 + Cr/0.6 + Mn/0.6 + V/1.4 + Fe/0.5 + Co/0.9 + Ni/0.8 (wt.%) [29], the Mo_eq_ values of the designed alloys increases monotonically with Mo content, and their values are 26.46~28.97%. This is largely well above the critical value (about 10 %) for a stable β solid solution [30]. It is noteworthy that although the design alloys have the same phase constitution, the diffraction peaks of β slightly shift towards bigger Bragg angles with Mo content. Compared with Ti and Zr, Mo has a smaller atomic radius. This inevitably lowers the lattice constant of the β solid solution, as shown in Table 2.

The typical microstructure of the designed alloys was exhibited in Figure 4. From the inset in Figure 4a, it can be clearly found that the Ti-Zr congruent alloy consists of long-striped α phases distributed along the boundary of β columnar crystals with epitaxial growth characteristics. Unlike the congruent alloy, the designed alloys are featured by near-equiaxed β crystals, showing a tendency of decreasing slightly in the grain size as Mo content increases (Figure 4b–f). Multi-field SEM observation reveals that the microstructure is very uniform in all designed alloys, but the microstructure between the substrate and the first deposited layer is an exception, where epitaxial columnar crystals are formed to replace the near-equiaxed crystals (Figure 5). Further EPMA analysis shows that the composition fluctuation is very small within each alloy, and the average composition is very close to the designed composition, which results from the strong convection stirring of the molten pool together with the high diffusion coefficients of Mo and Zr elements in the alloys [31]. As expected, the average content of Mo in the β grains increases when increasing Mo addition, whereas that of Zr shows an opposite trend as shown in Table 3, which is coincident with the lattice constant change of the β phase. However, at the interface between the substrate and the first deposited layer, there is an obvious increase in the content of Ti for all designed alloys, as a result of the dilution from the substrate. The detailed composition in this area is also listed in Table 3.

Grain orientation is a major parameter to characterize crystal structure, which directly affects the properties of alloys. Therefore, an EBSD analysis was performed. The orientation maps reveal that the β grain in all designed alloys exhibits morphological characteristics of near-equiaxed crystal (Figure 6a–e) and has a decrease in grain size as the Mo content increases (Table 4), which is consistent with the microstructure observed by SEM. In terms of the colors of the β grains, a rich variety of β grains can be discerned in all designed alloys, suggesting that their texture is weak and random. This can further be demonstrated by the pole figures of β grains, in which the pole distribution is promiscuous and irregular, and the pole density value is less than 10.

Grain boundary misorientation also has an important influence on mechanical properties since it acts as a stronger barrier to moving dislocations. From the histograms of grain boundary misorientation distribution in Figure 7, the grain boundaries in all designed alloys are mostly high-angle grain boundaries (HAGBs, defined as having a grain misorientation of >15° according to Brandon criterion) [32,33], and the distribution of HAGBs is close to the Mckenzie plot, indicating a random distribution of grain orientations. Further statistical analysis shows that the fraction of HAGBs increases from 86% to 91% with the increase in Mo content (Figure 7f).

Generally, an ultrafast cooling rate and steep temperature gradient of L-DED often induce the formation of columnar or dendritic grains with epitaxial growth characteristics. However, near-equiaxed grains are obtained in all designed alloys, which may be related to constitutional supercooling. According to the solidification theory, constitutional supercooling is caused by solute redistribution in the liquid-solid interface front. To evaluate the constitutional supercooling, a parameter of growth restriction factor Q, defined as the available amount of undercooling for the concentration of the initial solid to form is introduced, which could be expressed by ∑*m_i_c_i_*_0_ (*k_i_*−1), in which *m_i_* is the slope of the liquidus line, *c_i_*_0_ is the solute concentration (wt.%), *k_i_* is the solute partition coefficient, and *i* refers to individual solutes in the multi-component system [34]. Calculation represents that the Q values of the designed alloys linearly increase from 23.7 K to 52.5 K with the increase in Mo content (Figure 8), being much higher than the nucleation undercooling. To some extent, this could eliminate the harmful effects of the high-temperature gradient, which is favorable to the formation of near-equiaxed grains with multiple orientations [35,36]. However, the first deposited layers of all as-deposited alloys are an exception, where columnar grains with epitaxial growth characteristics are formed since high dilution from pure titanium substrate decreases Q (the value is 6.98~10.75 K), accompanied by a much higher temperature gradient. Moreover, the size of grains is inversely proportional to the Q based on interdependence theory, since an increased Q value will increase the nucleation rate, leading to the formation of more grains. This is the main reason why the size of the β phase reduces with Mo content. A similar result was also found in Zr-Nb-Ti alloys [37].

### 4.2. Mechanical Properties

The micro-hardness of the designed alloys as a function of the Mo content is shown in Figure 9. The micro-hardness was measured by employing a load of 10 N during an indent period of 15 s, and the average value of the forty-six different points of each alloy was taken as the final test data. The data reveal that the micro-hardness values increase with the addition of Mo because of the action of gradually refined microstructure and continuous solid solution enhancement. Their hardness values are HV443~HV475, which exceeds that (HV440) of the Ti-Zr congruent alloy.

Figure 10 illustrates the compressive stress-strain curves of the designed alloys at room temperature. Obviously, all designed alloys undergo three successive stages of elastic deformation, plastic deformation, and fracture during the compressive process. The mechanical properties after compression were listed in Table 5. It shows that both ultimate compressive strength and yield strength improve as the Mo content increases, accompanied by a decrease in the fracture strain. Compared with the mechanical properties of the Ti-Zr congruent alloy listed in Table 5, Mo addition effectively enhances the strength but decreases the ductility. Furthermore, the mechanical properties of the designed alloys are compared or superior to those of other titanium alloys fabricated by L-DED (their typical strength and ductility are 0.8–1.2 GPa and 1−24%, respectively) [7,38,39,40,41,42,43,44].

The fractographic observation shows that all designed alloys fracture along the direction of shear stress. At high magnification, all designed alloys show the typical ductile fracture features, with the fracture surface consisting of lots of dimples. Corresponding to the variation trend of the fracture strain, the size of the dimples gradually decreases with Mo content (Figure 11). This fracture mode results mainly from the BCC structure of β solid solution, which will produce high-density dislocations in it during compressive deformation. These dislocations easily move and propagate in the intragranular region without any obstacles such as the second phase. However, the slip transmission from one grain to another is difficult due to larger misorientation between grains, which will produce a bigger stress concentration and induces initiation and propagation of the microvoids in the grain boundary, leading to the fracture of the alloys through microvoid coalescence.

For single-phase solid solution alloys, their strengthening depends mainly on solid solution hardening and grain boundary strengthening. It is widely known that solid solution hardening results from the hindrance of solute atoms to dislocation slip. Relevant studies show that the total contribution from all alloying elements to solid solution hardening in multicomponent systems follows the approach of Gypen and Toda-Caraballo [45,46], which is given as:
$\sigma_{ss}={\left(\sum \beta_{i}^{3/2}x_{i}\right)}^{2/3}$, where *β_i_* is a constant that depends on crystal lattice and modulus mismatch of element *i* with solute, *x_i_* is the atomic percentage of the substitutional element *i* in a solid solution. In the current alloy system, the value of β_Mo_ is higher than that of β_Zr_, since the shear modulus difference between Mo and Ti is larger than that between Zr and Ti, and Mo reduces the lattice constant of β solid solution enlarged by Zr, as shown in Table 2. Therefore, increasing Mo content coupled with decreasing Zr content will enhance the solid solution hardening based on the above equation. In fact, grain boundary enhancement depends on the grain boundary’s blocking effect on the dislocation movement. The extent of the strengthening could be described by the Hall–Petch relation [47,48]: *σ*_HP_ = *k*_HP_ / √*D*, where *D* is the mean grain size and *k_HP_* is Hall–Petch constant. Therefore, increasing the Mo content decreases the size of the grain and dislocation movement was restricted by more grain boundaries, resulting in progressively enhanced grain boundary strengthening. What is more, the fraction of LAGBs also increases with the addition of Mo content (Figure 7), which will increase the stress concentration at grain boundaries, being conducive to the improvement in strength. In summary, the strength of the designed alloys improves gradually with the increase in Mo content resulting from the joint action of various strengthening mechanisms but have a reduced ductility due to the decreased plastic flow caused by enhanced solid solution hardening and increased LAGBs fraction, even though the gradual refinement of grains favors the improvement of ductility.

### 4.3. Tribological Properties

Figure 12 displays the effect of the Mo content on the tribological properties of the designed alloys. Clearly, the worn volume and friction coefficient reduces gradually as Mo content increases, which suggests that increasing the Mo content can effectively improve the anti-friction property and the wear resistance. Moreover, the tribological properties of the designed alloys are superior to those of the Ti-Zr congruent alloy, whose worn volume is 0.2884 mm^3^ and friction coefficient is 0.590.

To further investigate the wear mechanism, the worn morphologies of the surface of the designed alloy were observed by SEM. Plowing grooves that characterize abrasive wear occur on the surface of the designed alloys, and become narrow and shallow when increasing Mo content (Figure 13) indicating a gradual enhancement in the anti-abrasive wearability. Based on the principles of tribology, abrasive wear appears on a soft metal surface abraded by hard abrasive particles. Depending on different conditions, several mechanisms have been proposed, including the micro-cutting mechanism, micro-fracture mechanism, and fatigue failure mechanism. From the worn surface characteristics, one can infer that the micro-cutting mechanism is dominant. The force component which is normal to the surface and acts on the Si_3_N_4_ grinding ball causes the penetration of the surface by the ball. That force which is parallel to the surface causes relative tangential motion between the surface and the ball. During the execution of the process, the surface material will be pushed transversely to the direction of the ball motion to form grooves. Most displaced material piles up along the groove edge rather than being removed from the surface. The severity of the wear by this mechanism greatly depends on the hardness of alloys, that is to say, high hardness will decrease the cutting efficiency of abrasives, resulting in a reduction in the wear rates. Therefore, the hardness is closely related to the tribological properties of the designed alloys.

### 4.4. Corrosion Resistance

The measurement of the potential dynamic polarization test on the designed alloys was carried out in an HCl solution. From anodic polarization curves (Figure 14), all designed alloys experience a similar corrosion process: the initial current density rises dramatically as the corrosion potential increases due to the active dissolution of the designed alloys. As the potential goes beyond a specific threshold, the current density approximately remains constant with the rapid shift in the potential towards positive, meaning that the designed alloys appear stable passivation films. This is followed by over-passivation and secondary passivation. This may be due to the formation of high-valence ions by metal dissolution or other ions participating in the anodic reaction. According to the Tafel linear extrapolation method, the self-corrosion potential (Ecorr) and self-corrosion current density (Icorr) of the as-deposited Ti-Zr-Mo alloys were obtained by fitting and shown in Table 6. Results show that the Ecorr shifts towards a noble value as Mo content increases, while the Icorr gradually decreases. Thus, the corrosion resistance is an increasing function of Mo content and shows an outstanding advantage over Ti-Zr congruent alloys. Besides, Ecorr is −0.2 VSCE and Icorr is 9.3 × 10^−8^ A/cm^2^.

Figure 15 presents the corroded surface morphologies of the designed alloys observed under SEM. Owing to the microstructure feature of a single β solid solution, the surface of the alloys presents morphological characteristics of uniform corrosion, on which a continuous and complete passivation film is formed (Figure 15a–e). This differs from the Ti-Zr congruent alloy, on which the eroding pits distributed along grain boundaries are clearly visible, resulting from galvanic micro-cells generated by the dual phase structure of α and β. In an HCl solution, the α phase has a lower potential than the β phase, thus the former serves as the anode with respect to the latter. As a result, preferential corrosion begins from the α phase, which leads to selective corrosion, leaving eroding pits along grain boundaries.

To further reveal the corrosion mechanism, the chemical composition of the passive films was tested by XPS. During the process of deconvolution fitting, the high-resolution peaks can be obtained by a non-linear least-squares algorithm with a Shirley baseline and a Gaussian–Lorentzian combination which are Ti 2p, Zr 3d, Mo 3d, and O 1s (Figure 16). The Ti 2p spectrum is composed of two peaks at 458.1 eV and 463.9 eV, which are assigned to the Ti^4+^ state. The Zr 3d spectrum appears as two peaks at 181.8 eV and 184.1eV, which are identified as Zr^4+^ state. The Mo 3d spectrum does not exhibit prominent characteristic peaks at 0.78 at.% Mo. However, when the Mo content is beyond the limit, two peaks corresponding to Mo^5+^ at 231.25 eV and 234.45 eV are clearly detected and show gradually enhanced intensity with the further increase in Mo content. The O 1s spectrum is fitted with two peaks at 530 eV and 532 eV, which denote O^2-^ and OH^−^, respectively. The O^2−^ peak is more intense than the OH^−^. Therefore, one can conclude that all passive films mainly consist of TiO_2_, ZrO_2_, and Mo_2_O_5_, among which Mo_2_O_5_ has an increase in its fraction when increasing Mo content.

The corrosion resistance of the alloys depends strongly on the structure, stability and compactness of passive films. Based on the above analysis, one can know that the passive films of all designed alloys contain Ti^4+^, Zr^2+^, and Mo^5+^ interstitials or oxygen vacancies. When the surface of the deposited alloy is in contact with the solution, it dissolves and then the cation interstitials are transferred to the solution where they bind to the oxygen ions to form TiO_2_, ZrO_2_, and Mo_2_O_5_ oxides. TiO_2_ oxide is helpful to form stable passivation films in acidic solutions. The existence of a certain amount of ZrO_2_ can further increase the thermodynamic or kinetic stability of TiO_2_ and thus reduce the dissolution rate of TiO_2_ oxides since the Zr^4+^-O bond has higher formation energy than the Ti^4+^-O bond. However, too much ZrO_2_ could cause the phenomenon of transpassivation, because it is sensitive to reactive chloride ions. When the Mo element is oxidized and acts as a hybrid vacancy, it can effectively increase the stability and compactness of passive films. Moreover, the size of the grains can lead to a profound influence on the corrosion resistance of the designed alloys. The electron activity and the electrochemical reactivity at the grain boundary would get improved by grain refinement, which will promote the rapid formation of passive films, protecting the alloys and enhancing corrosion resistance [49]. Taken together, the increased corrosion resistance with Mo content is mainly due to the increased stability of passive films and the constantly refined grains [50].

### 4.5. Density

To evaluate the internal quality of the designed alloys, four cross-sections of each alloy were sectioned along the built direction and made into metallographic specimens through grinding and polishing processes. It could be seen that no inadequate fusion and micro-cracks appear in all cases, but gas pores with the size of 10~500 μm are clearly visible. Such typical micrographs are displayed in Figure 17. More detailed statistics exhibit that the average porosity of the designed alloys gradually increases as the Mo content increases, the values are in the range of 0.11–0.68%, higher than (0.06%) without Mo addition.

The gas pore is a common metallurgical defect in L-DEDed titanium alloys. It is generally believed that the generation of gas pores is associated with the gas trapped in the melt pool. During the fast cooling process of L-DED, gas cannot escape in time and is left to form pores. Feedstock powders are a vital factor for the porosity in as-deposited alloys. In general, spherical powders have less chance of forming gas pores than irregular powders since the latter often contain a certain level of porosity, which will offer a great opportunity that the gas inside and entrained with the powder particles to be trapped in the melt pool. In current practice, the Zr and Mo powders were irregular hydride-dihydride powders that contain many gas pores. Although the powder mixture was dried for a long time in a vacuum drying oven, little oxygen was still detected in it and showed an increase in its amount with the increase in Mo content (Figure 18). This inevitably increases the porosity of the designed alloys at the same flow rate as the shield gas. The convective effect of the molten pool is another important factor affecting the porosity of the designed alloys. The enhanced dynamics of the flow can suppress the external pressure of gas bubble initiation to a great extent, which decreases the saturation of gas in liquid metal and increases the rate of gas escape so as to effectively reduce the porosity of the designed alloys. Based on this, JmatPro was used to quantitatively simulate the thermal conductivity of as-deposited Ti-Zr-Mo alloys. The corresponding results show that increasing Mo content raises the thermal conductivity from 26.71 w/m·K to 27.69 w/m·K. This implies a decrease in the temperature gradient of the molten pools, which in turn reduces the melt pool’s surface tension, finally weakening the intensity of convection. Therefore, it can be concluded that the feedstock powder and the convective effect are essential for the porosities of the as-deposited Ti-Zr-Mo alloys. Both the denser powder and the enhanced convective effect will help to reduce the porosity of the as-deposited alloys.

### 4.6. Surface Roughness

The surface roughness could be a key parameter to measure the forming performance of as-deposited alloys. In the initial measurements, the top roughness of alloys is more representative owing to the slightly greater roughness than that of the side walls. Consequently, a roughness tester measured three-dimensional profiles from top surfaces for all as-deposited alloys. The results indicate that the surface roughness has a clear dependence on Mo content, namely, increasing Mo content increases the Ra value from 1.13 μm to 2.14 μm (Figure 19). Compared with the Ra value (0.47 μm) of the Ti-Zr congruent alloy, those of the designed alloys are much higher, which suggests that Mo addition decreases the formability.

A variety of mechanisms could contribute to the roughness such as the alloy composition, laser processing parameters, and the complex thermodynamic changes of metal powders from melting to solidification. Under the optimized process parameters, the alloy compositions affect greatly the L-DED surface roughness, because the fluidity or spread-ability of the melt varies with the composition. This will directly affect the surface finish of the Ti-Zr-Mo alloys. Guided by the theory of metal solidification, the liquid-state fluidity relies heavily on the solidification interval of the alloy. Generally speaking, the narrower the solidification interval is, the better the liquid-state fluidity of the alloy is [51]. Thus, JmatPro was used to quantitatively simulate the solidification temperature range of all as-deposited alloys. When Mo content increases, the solidification temperature interval broadens gradually shown in Figure 20, which causes a tendency to decrease the surface finish of the designed alloys. A stair-stepping effect, in the process of stacking layer by layer, is also an important source of surface roughness. To weaken the stair-stepping effect and improve the surface finish, a good spread-ability of melt is required. Therefore, the spread-ability was evaluated through a single-track cladding experiment. The result shows that increasing Mo content leads to a linear decrease in the aspect ratio of the cladding layers (Figure 21), which implies that the spread-ability of the melt becomes worse. Under the combined influence of the above factors, the surface roughness increases gradually with Mo content.

## 5. Conclusions

In this work, a new series of Ti-Zr-Mo alloys containing different Mo contents were designed using an advanced cluster model and then made by L-DED on a pure titanium substrate. Without any special process control, near-equiaxed β grains without obvious texture were obtained for all designed alloys. The mechanism of the microstructure formation was analyzed using the growth restriction factor. A systematic investigation was performed on mechanical, tribological, chemical and forming properties of the designed alloys to throw light on how the microstructural features impact these properties. The designed alloys exhibit a superior combination of strength, wear and corrosion resistances, compared to Ti-Zr congruent alloys in a similar processing technology. The following conclusions are drawn:
(1).The microstructural characteristics reveal that all designed alloys are made up of near-equiaxed β grains without obvious texture, which results mainly from high growth restriction factor with different alloys. When increasing Mo content, the β grain is gradually refined and its lattice constant is steadily reduced.(2).Increasing Mo content can effectively improve the hardness, strength, and wear resistance of the designed alloys due to the refinement of the grain and enhancement of solid solution hardening, but reduce ductility owing to the combination of the decreased plastic flow and increased LAGBs fraction. Under dry sliding friction and wear conditions, the main friction mechanism of the designed alloys is abrasive wear, the increasing hardness with Mo content improves the wear resistance of the designed alloys.(3).The corrosion resistance of the designed alloys in HCl solution is mainly governed by Mo content through the introduction of more Mo_2_O_5_ oxides in the passive film that effectively increases the stability of passive films, leading to enhanced corrosion resistance.(4).The average porosity and surface roughness of designed alloys in the process of Mo content increases, which is aroused mainly by the interplay of various factors, including raw powder density, convection intensity, fluidity and spreadability of melt.(5).The Ti_60.94_Zr_36.72_Mo_2.34_ alloy can be considered a promising candidate for L-DED, resulting from a good match of mechanical, tribological, chemical, and forming properties.


## Figures and Tables

**Figure 1 materials-16-01054-f001:**
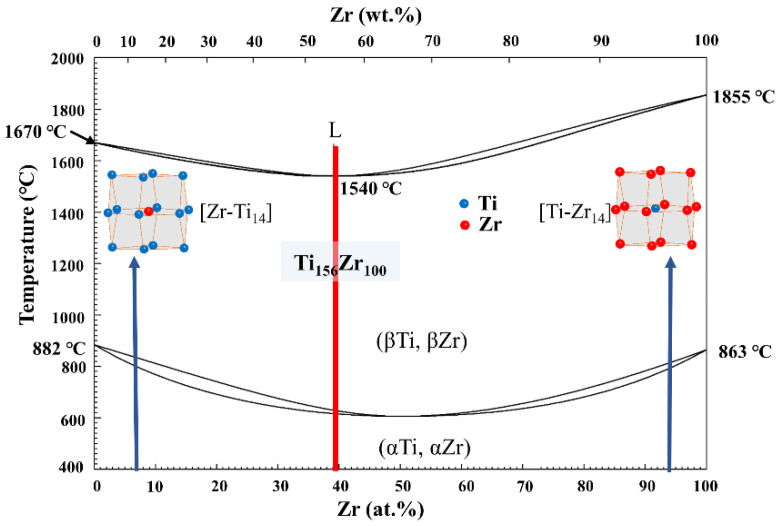
Ti-Zr binary phase diagram and cluster structure.

**Figure 2 materials-16-01054-f002:**
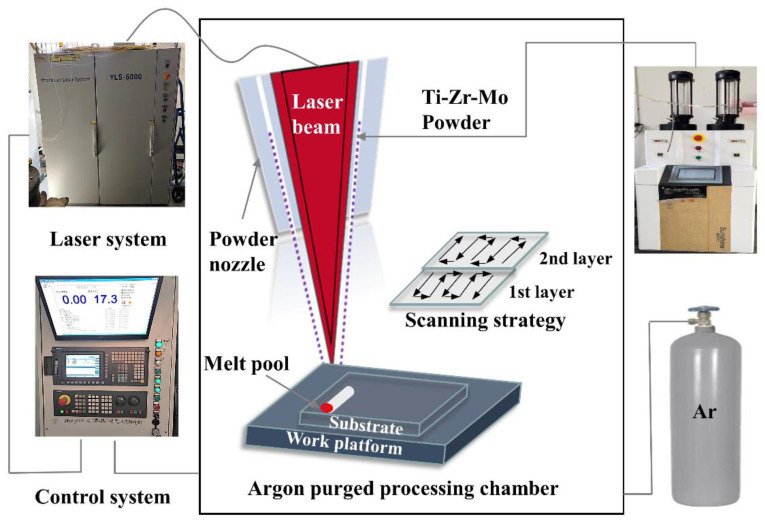
The system of laser direct energy deposition system.

**Figure 3 materials-16-01054-f003:**
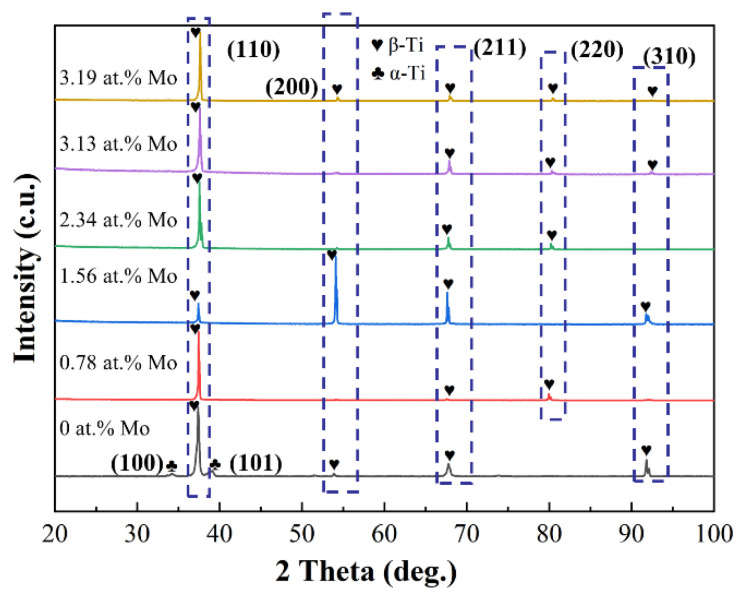
XRD patterns of the designed alloys.

**Figure 4 materials-16-01054-f004:**
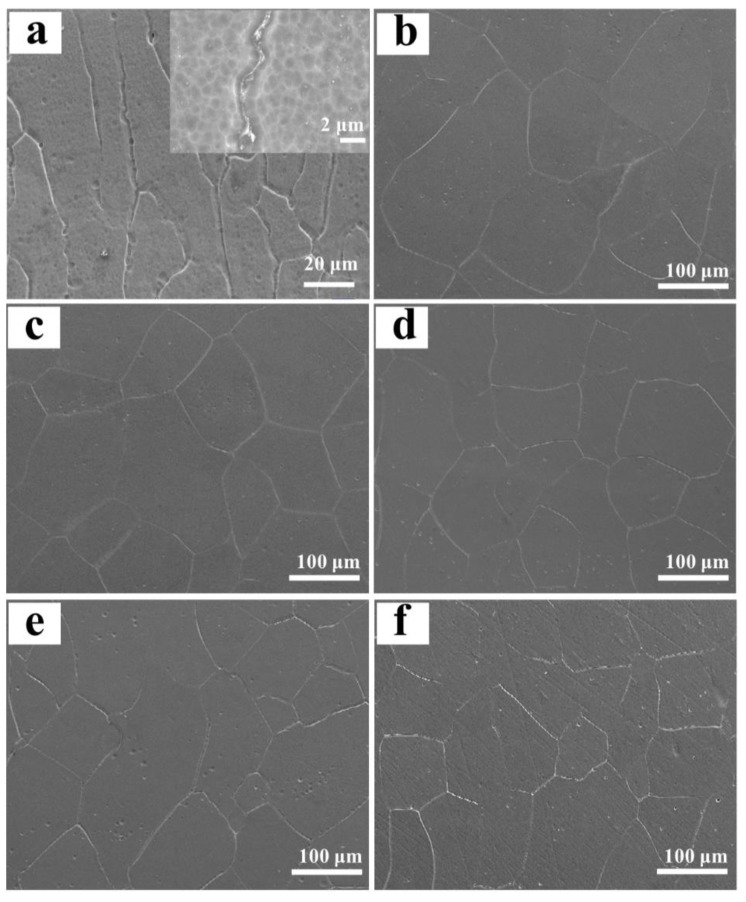
Typical SEM morphologies of the Ti-Zr congruent alloy and designed alloys: (**a**) Ti-Zr congruent alloy; (**b**) 0.78 at.% Mo; (**c**) 1.56 at.% Mo; (**d**) 2.34 at.% Mo; (**e**) 3.13 at.% Mo; (**f**) 3.91 at.% Mo.

**Figure 5 materials-16-01054-f005:**
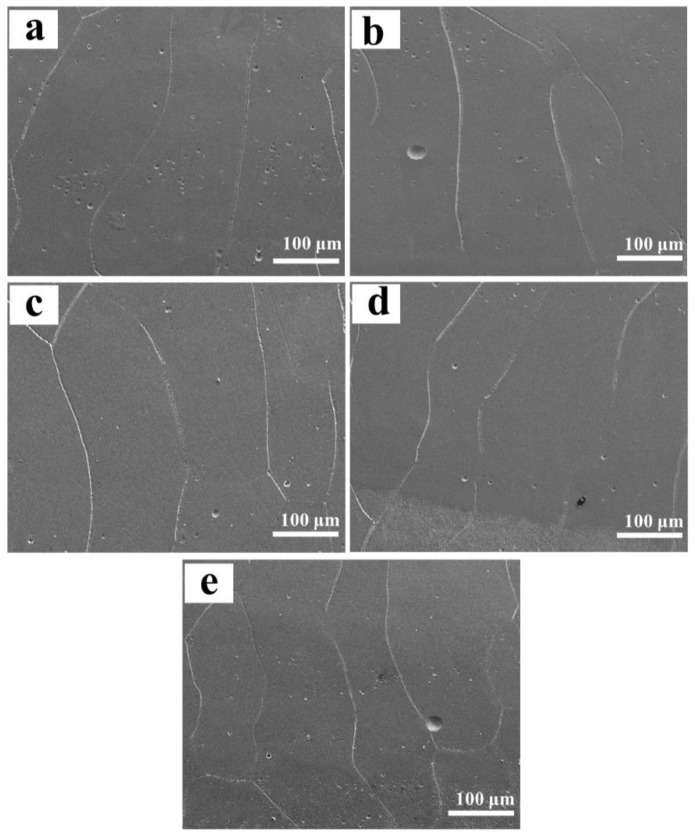
SEM morphologies taken from the interface of the first deposited layer and the substrate: (**a**) 0.78 at.% Mo; (**b**) 1.56 at.% Mo; (**c**) 2.34 at.% Mo; (**d**) 3.13 at.% M (**e**) 3.91 at.% Mo.

**Figure 6 materials-16-01054-f006:**
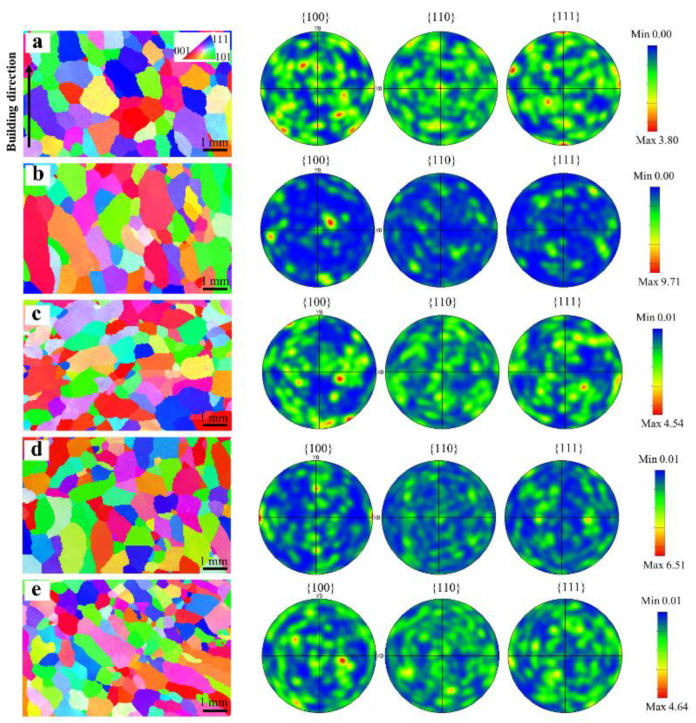
EBSD orientation maps and pole figures of the designed alloys: (**a**) 0.78 at.% Mo; (**b**) 1.56 at.% Mo; (**c**) 2.34 at.% Mo; (**d**) 3.13 at. % Mo; (**e**) 3.91 at.% Mo.

**Figure 7 materials-16-01054-f007:**
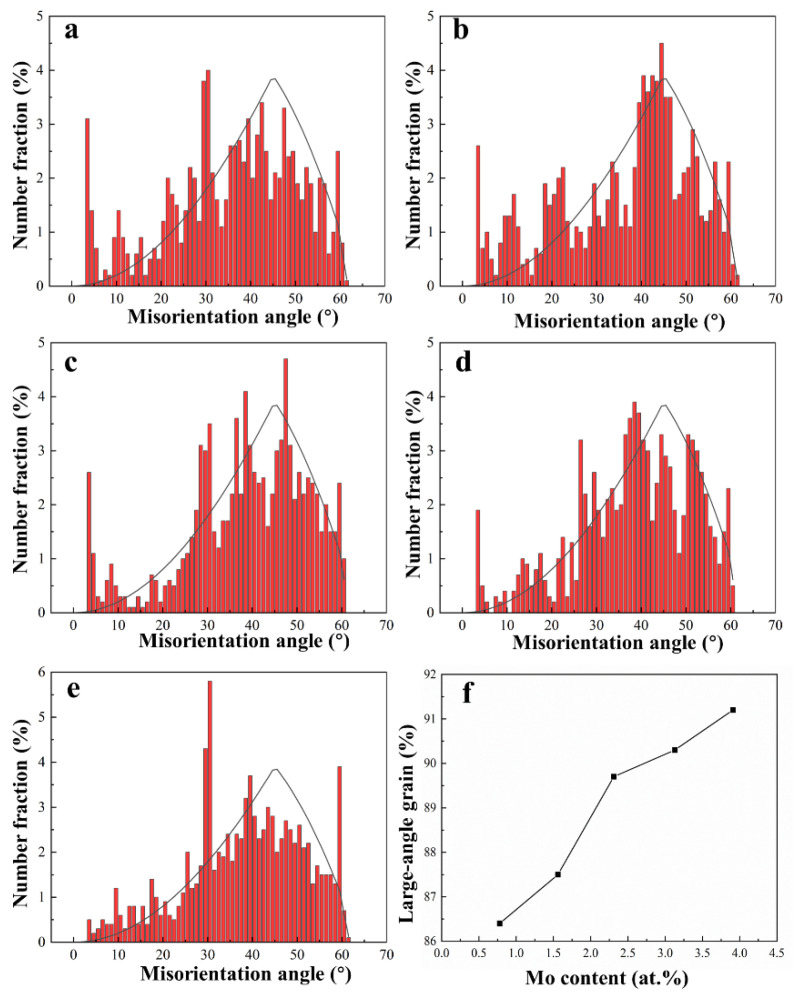
Misorientation distribution maps of the designed alloys: (**a**) 0.78 at.% Mo; (**b**) 1.56 at.% Mo; (**c**) 2.34 at.% Mo; (**d**) 3.13 at.% Mo; (**e**) 3.91 at.% Mo; (**f**) Large-angle grain boundary fraction.

**Figure 8 materials-16-01054-f008:**
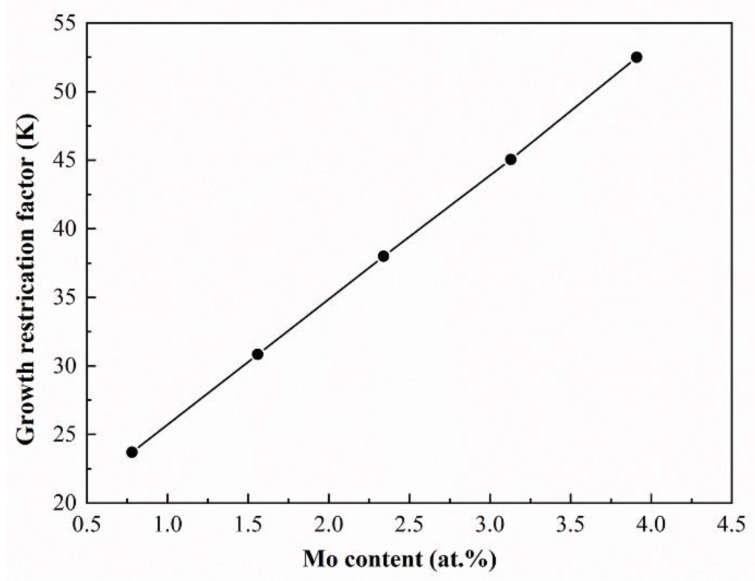
The growth restriction factors of the designed alloys.

**Figure 9 materials-16-01054-f009:**
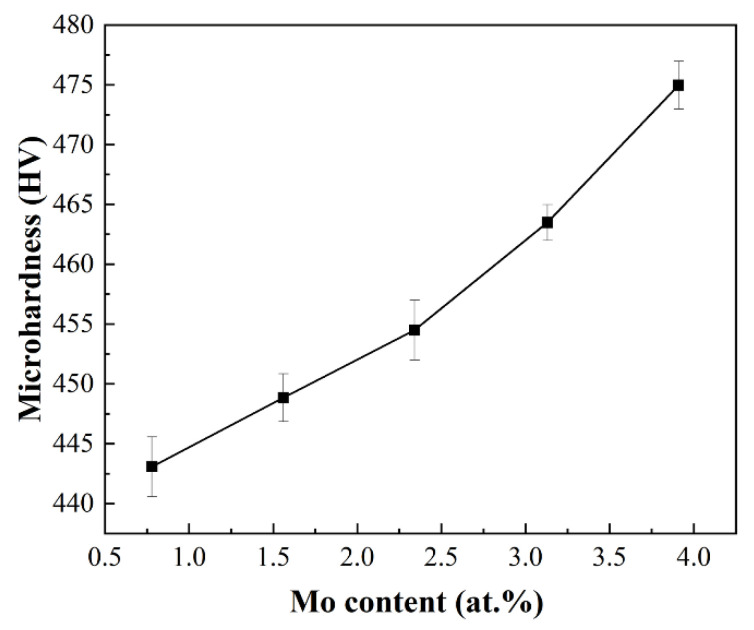
The influence of Mo content on average micro-hardness of the designed alloys.

**Figure 10 materials-16-01054-f010:**
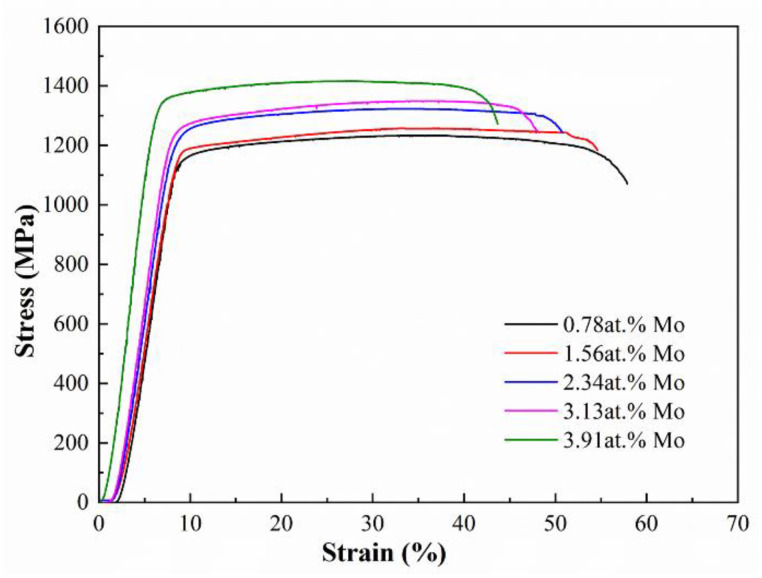
The stress-strain curves of the designed alloys at room temperature.

**Figure 11 materials-16-01054-f011:**
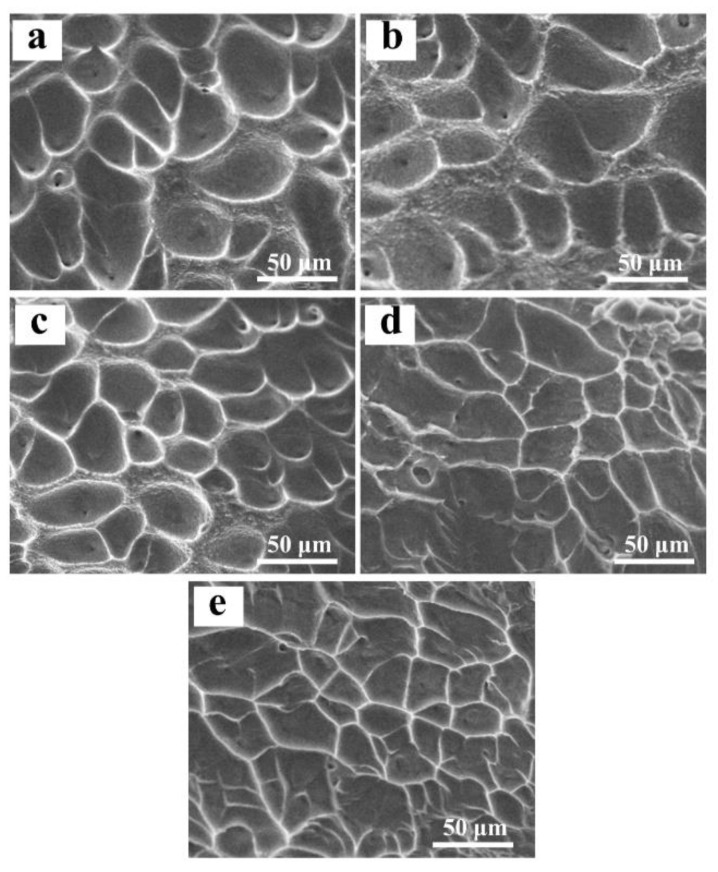
Fracture morphologies of the designed alloys: (**a**) 0.78 at.% Mo; (**b**) 1.56 at.% Mo; (**c**) 2.34 at.% Mo; (**d**) 3.13 at.% Mo; (**e**) 3.91 at.% Mo.

**Figure 12 materials-16-01054-f012:**
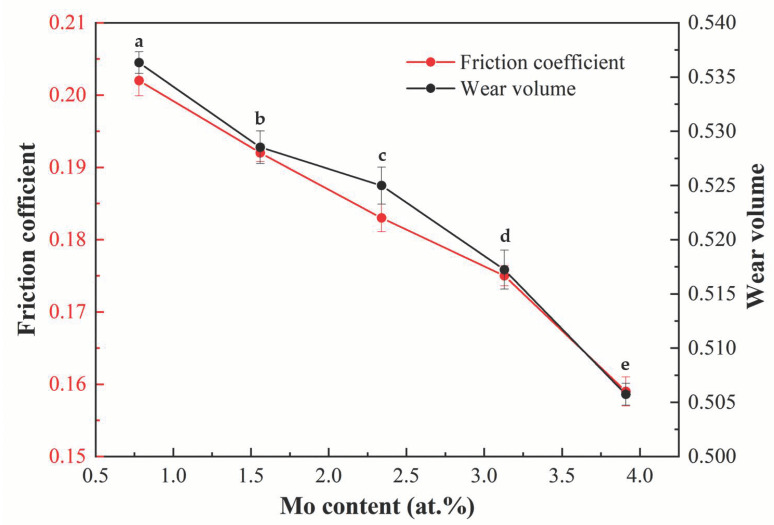
The effect Mo content on the tribological properties of the designed alloys: (a) 0.78 at.% Mo; (b) 1.56 at.% Mo; (c) 2.34 at.% Mo; (d 3.13 at.% Mo; (e) 3.91 at.% Mo.

**Figure 13 materials-16-01054-f013:**
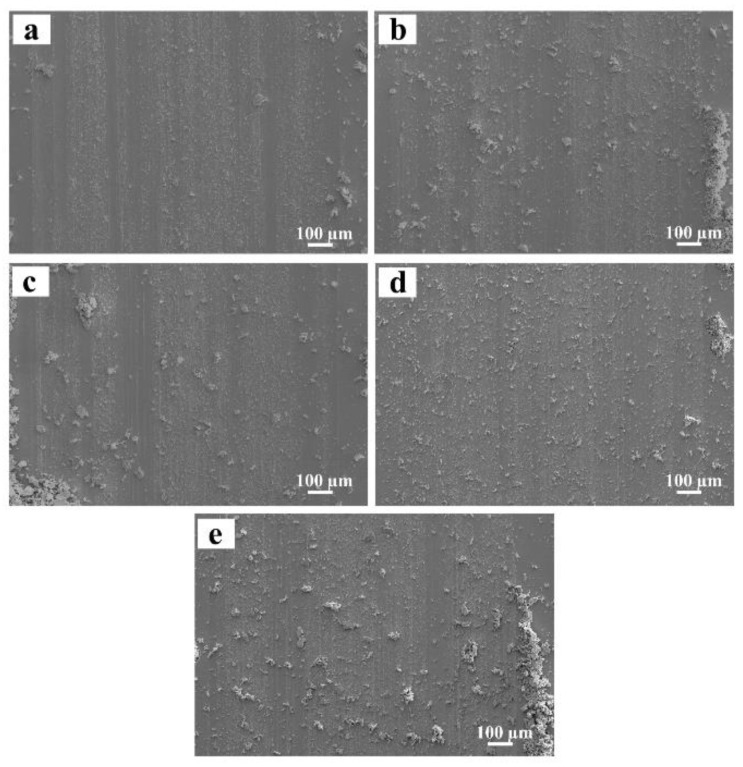
The worn morphologies of the designed alloys: (**a**) 0.78 at.% Mo; (**b**) 1.56 at.% Mo; (**c**) 2.34 at.% Mo; (**d**) 3.13 at.% Mo; (**e**) 3.91 at.% Mo.

**Figure 14 materials-16-01054-f014:**
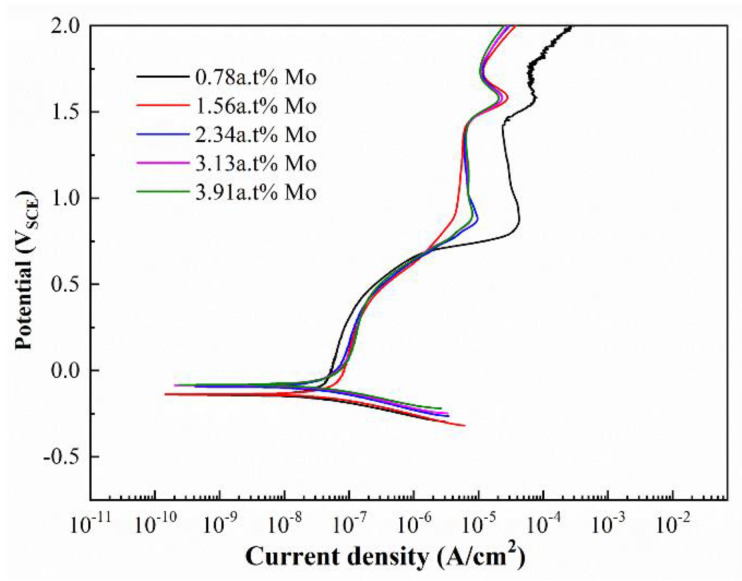
Potentiodynamic polarization curves of the designed alloys.

**Figure 15 materials-16-01054-f015:**
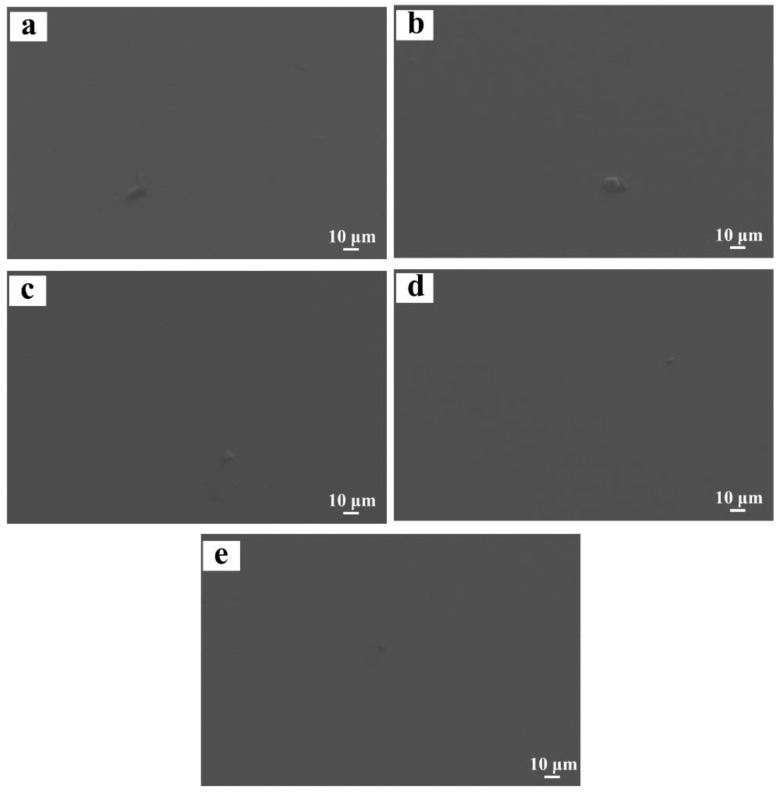
Corroded surface morphologies of the designed alloys: (**a**) 0.78 at.% Mo; (**b**) 1.56 at.% Mo; (**c**) 2.34 at.% Mo; (**d**) 3.13 at.% Mo; (**e**) 3.91 at.% Mo.

**Figure 16 materials-16-01054-f016:**
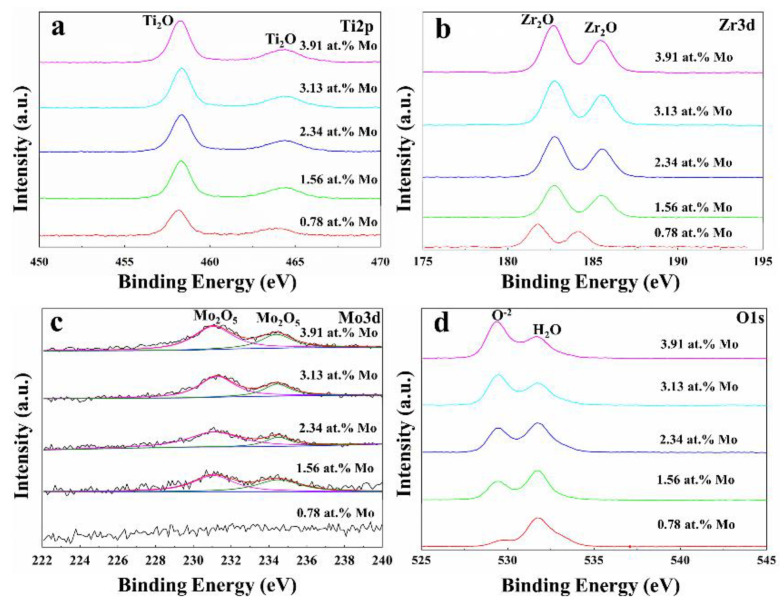
XPS high-resolution spectrums of the designed alloys: (**a**) Ti 2p; (**b**) Zr 3d; (**c**) Mo 3d; (**d**) O 1s.

**Figure 17 materials-16-01054-f017:**
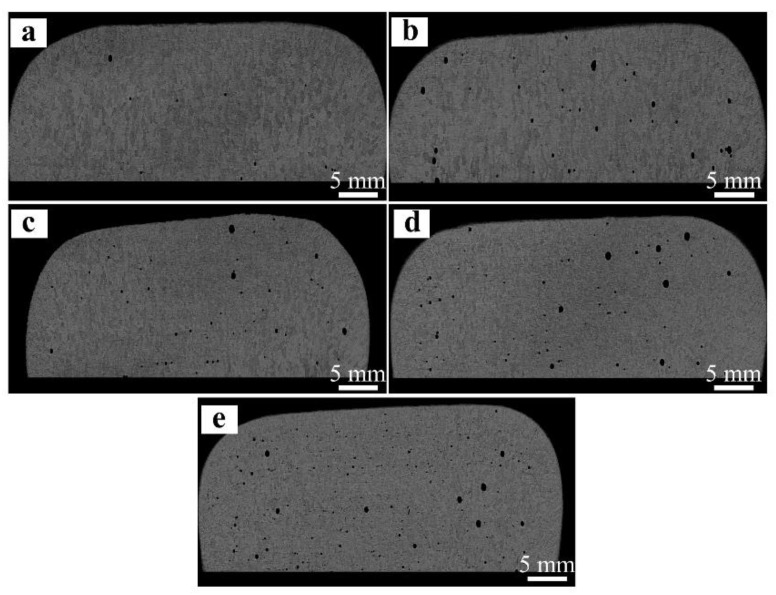
Optical metallographic photographs taken from the cross-sections of the designed alloys: (**a**) 0.78 at.% Mo; (**b**) 1.56 at.% Mo; (**c**) 2.34 at.% Mo; (**d**) 3.13 at.% Mo; (**e**) 3.91 at.% Mo.

**Figure 18 materials-16-01054-f018:**
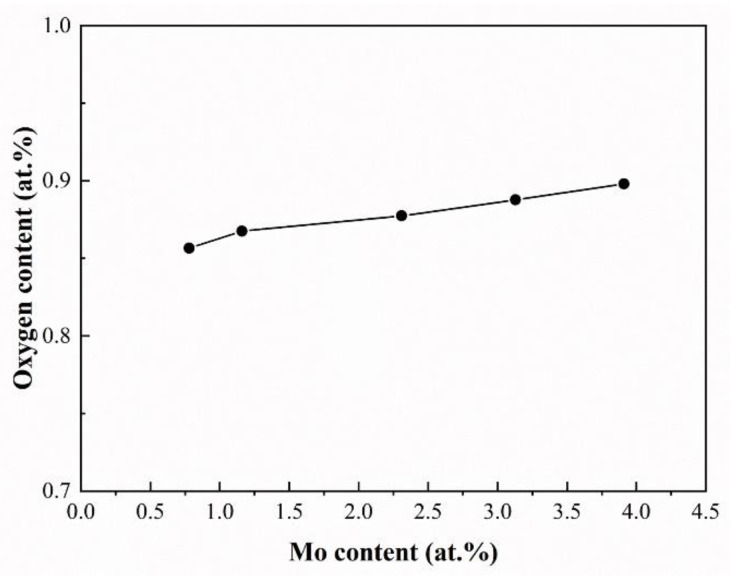
The effect of Mo content on the oxygen content of the raw mixed powders.

**Figure 19 materials-16-01054-f019:**
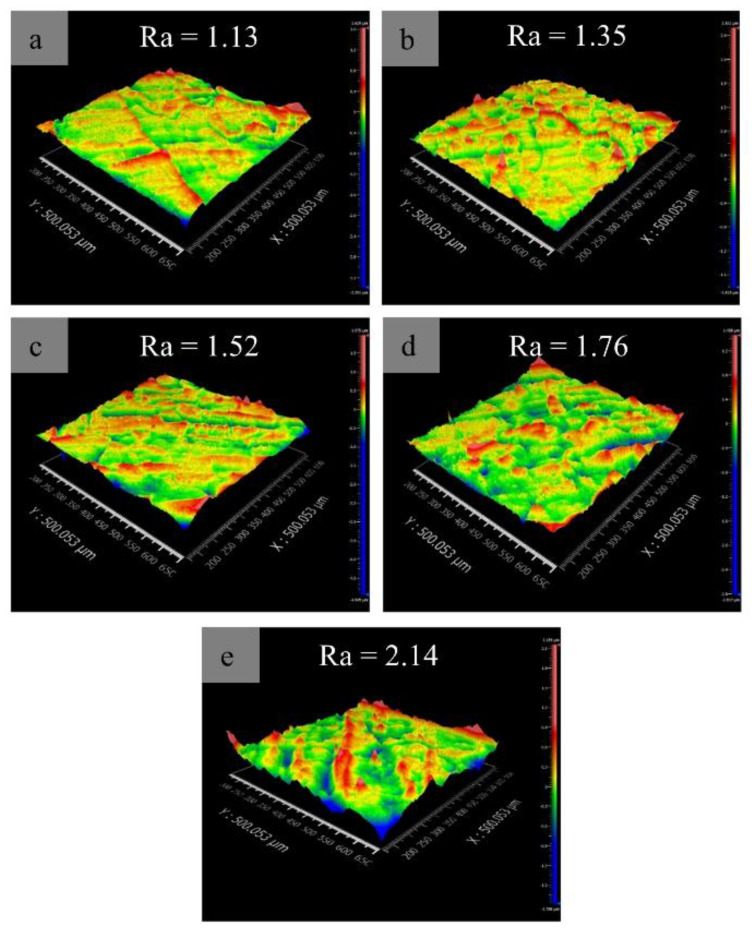
Typical 3D outline drawing taken from the surfaces of the designed alloys: (**a**) 0.78 at.% Mo; (**b**) 1.56 at.% Mo; (**c**) 2.34 at.% Mo; (**d**) 3.13 at.% Mo; (**e**) 3.91 at.% Mo.

**Figure 20 materials-16-01054-f020:**
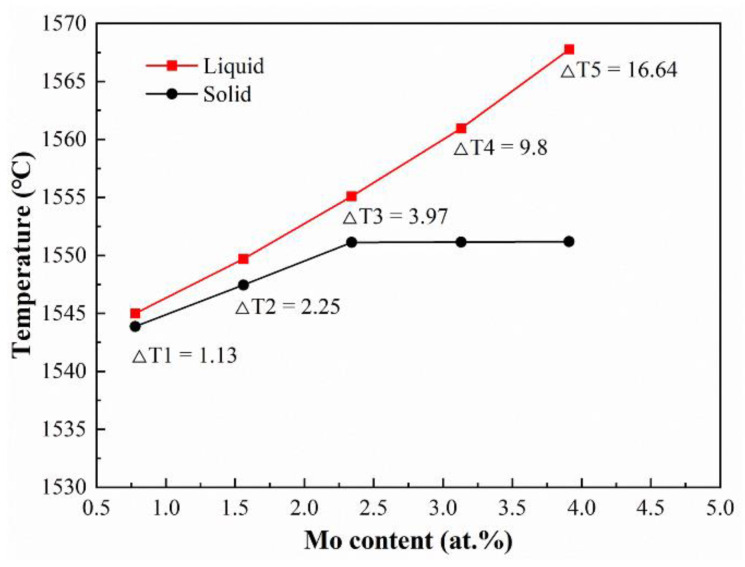
The influence of Mo content on the liquid-solid temperature of the designed alloys.

**Figure 21 materials-16-01054-f021:**
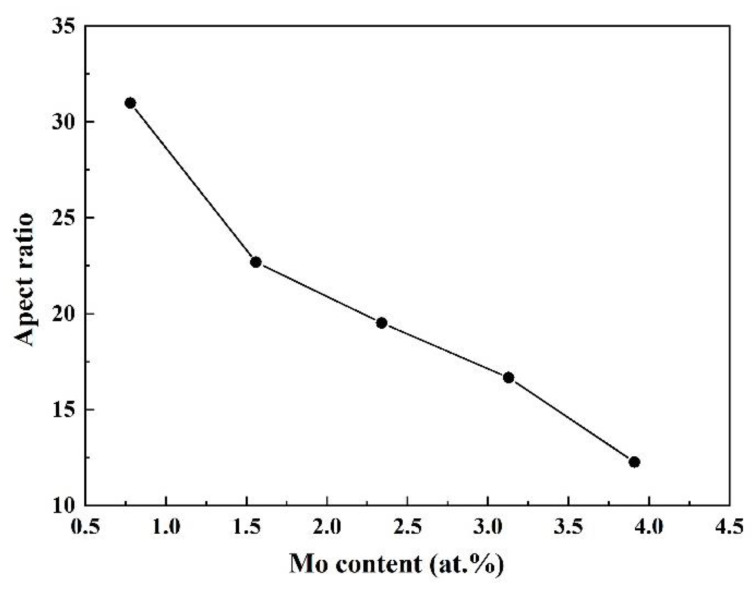
The aspect ratios of single-track deposited layers with various Mo contents.

**Table 1 materials-16-01054-t001:** Correspondence between cluster formula and composition of Ti-Zr-Mo alloys.

Mo Content (at.%)	Cluster Formula	Composition (at.%)
0.78	10[(Zr_0.8_Mo_0.2_)-Ti_14_]Ti_1_ + 6[Ti-Zr_14_]Zr_1_	Ti_60.94_Zr_38.28_Mo_0.78_
1.56	10[(Zr_0.6_Mo_0.4_)-Ti_14_]Ti_1_ + 6[Ti-Zr_14_]Zr_1_	Ti_60.94_Zr_37.5_Mo_1.56_
2.34	10[(Zr_0.4_Mo_0.6_)-Ti_14_]Ti_1_ + 6[Ti-Zr_14_]Zr_1_	Ti_60.94_Zr_36.72_Mo_2.34_
3.13	10[(Zr_0.2_Mo_0.8_)-Ti_14_]Ti_1_ + 6[Ti-Zr_14_]Zr_1_	Ti_60.94_Zr_35.93_Mo_3.13_
3.91	10[(Mo_1_)-Ti_14_]Ti_1_ + 6[Ti-Zr_14_]Zr_1_	Ti_60.94_Zr_35.15_Mo_3.91_

**Table 2 materials-16-01054-t002:** Lattice constant of β solid solution in the designed alloys.

Alloy	Lattice Parameter (nm)
Ti_60.94_Zr_38.28_Mo_0.78_	0.3393
Ti_60.94_Zr_37.5_Mo_1.56_	0.3385
Ti_60.94_Zr_36.72_Mo_2.34_	0.3384
Ti_60.94_Zr_35.93_Mo_3.13_	0.3379
Ti_60.94_Zr_35.15_Mo_3.91_	0.3374

**Table 3 materials-16-01054-t003:** Average chemical composition of β solid solution from designed alloys.

Alloy	Average Chemical Composition (at.%)					
	Major Deposited Layers			Interface Zone of the First Deposited Layer and the Substrate		
	Ti	Zr	Mo	Ti	Zr	Mo
Ti_60.94_Zr_38.28_Mo_0.78_	61.36	37.90	0.74	66.18	33.21	0.61
Ti_60.94_Zr_37.5_Mo_1.56_	60.78	37.64	1.58	67.05	31.51	1.44
Ti_60.94_Zr_36.72_Mo_2.34_	61.16	36.56	2.28	67.61	30.32	2.07
Ti_60.94_Zr_35.93_Mo_3.13_	61.02	35.71	3.27	67.89	28.99	3.12
Ti_60.94_Zr_35.15_Mo_3.91_	60.59	35.43	3.98	68.01	28.46	3.53

**Table 4 materials-16-01054-t004:** The average size of β grains in the designed alloys.

Alloy	Grain Size (µm)
Ti_60.94_Zr_38.28_Mo_0.78_	191.61
Ti_60.94_Zr_37.5_Mo_1.56_	183.34
Ti_60.94_Zr_36.72_Mo_2.34_	180.45
Ti_60.94_Zr_35.93_Mo_3.13_	177.31
Ti_60.94_Zr_35.15_Mo_3.91_	159.06

**Table 5 materials-16-01054-t005:** Compressive test data of the designed alloys.

Alloys (at.%)	Yield Strength (MPa)	Ultimate Compressive Strength (MPa)	Relative Compressibility (%)
Ti_60.94_Zr_38.28_Mo_0.78_	1232.98	1087.52	45.20
Ti_60.94_Zr_37.5_Mo_1.56_	1257.30	1094.54	42.75
Ti_60.94_Zr_36.72_Mo_2.34_	1323.05	1132.95	40.27
Ti_60.94_Zr_35.93_Mo_3.13_	1349.08	1196.97	38.25
Ti_60.94_Zr_35.15_Mo_3.91_	1415.92	1269.75	36.32
Ti-Zr congruent alloy	813.00	970.00	57.00

**Table 6 materials-16-01054-t006:** Electrochemical properties of the designed Ti-Zr-Mo alloys.

Alloys	Ecorr (V/SCE)	Icorr (A/cm^2^)
Ti_60.94_Zr_38.28_Mo_0.78_	−0.141	6.090 × 10^−8^
Ti_60.94_Zr_37.5_Mo_1.56_	−0.137	5.412 × 10^−8^
Ti_60.94_Zr_36.72_Mo_2.34_	−0.093	5.085 × 10^−8^
Ti_60.94_Zr_35.93_Mo_3.13_	−0.084	4.202 × 10^−8^
Ti_60.94_Zr_35.15_Mo_3.91_	−0.082	4.198 × 10^−8^

## Data Availability

Not applicable.

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
