# Peer review of "Microstructure and Properties of Ti-Zr-Mo Alloys Fabricated by Laser Directed Energy Deposition"

_materials, 2023, doi:10.3390/ma16031054_

Round 1

Reviewer 1 Report

1. The authors must explain the novelty of the current work.

2. Is it right to call ' congruent alloys' ? Please support with references.

3. Expand and explain TA2 substrates.

4. Figure 3. X-ray diffraction patterns of as-deposited Ti-Zr-Mo alloys. Peaks have to be indexed for their respective crystallographic planes.

5.What is the significance of  growth restriction factor?

6. Figure 9. Mention the load used in microhardness.

7. How did the authors observe fracture morphologies when got stress strain curves by compression tests?

8. There is no strong conclusion on the wear mechanism obtained.

9. The plagiarism to be reduced to less than 15%. Report attached. 

Author Response

Dear Reviewer

Thank you for your useful comments and suggestions. We have modified the manuscript accordingly, and detailed corrections are listed below:

1. The authors must explain the novelty of the current work.

The novelty of the current work has been explained in conclusion section.

2. Is it right to call 'congruent alloys'? Please support with references.

The term for congruent alloy is correct, and relevant reference has been added.

3. Expand and explain TA2 substrates.

In fact, the substrate material used for L-DED is high purity titanium with mass purity of 99.90% rather than commercial pure titanium TA2. This error has been corrected in revised manuscript.  

4. Figure 3. X-ray diffraction patterns of as-deposited Ti-Zr-Mo alloys. Peaks have to be indexed for their respective crystallographic planes.

According to the comment, the diffraction peaks have been indexed for their respective crystallographic planes.

5. What is the significance of growth restriction factor?

The significance of growth restriction factor has been added.

6. Figure 9. Mention the load used in micro-hardness.

The load used in micro-hardness test has been added.

7. How did the authors observe fracture morphologies when got stress strain curves by compression tests?

After the compressive deformation, the fracture morphologies were observed by SEM to investigate the fracture mechanism of the as-deposited alloys, which has been added to experimental procedures section.

8. There is no strong conclusion on the wear mechanism obtained.

The conclusion on the wear mechanism has been added.

9. The plagiarism to be reduced to less than 15%. Report attached.

Thanks for careful review. The manuscript has been carefully revised, and the repetition rate has been reduced to below 15%.

Reviewer 2 Report

The article "Microstructure and properties of Ti-Zr-Mo alloys fabricated by laser directed energy deposition" is interesting, very well written and the topic addressed is a subject of interest for both the mechanical and medical fields. I recommend accepting the article in its present form for publication.

More comments are attached

Author Response

1.Potential practical applications that could use the Ti60.94Zr36.72Mo2.34 alloy (at.%) are not detailed at all.

The designed alloys are widely used in the aeroengine components.

2.By considering all these strength points & weaknesses, I recomend to accept the manuscript  Microstructure and properties of Ti-Zr-Mo alloys fabricated by laser directed energy deposition after minor revision (corrections to minor methodological errors and text editing).

Thank for your constructive comments, I have finished the corrections to minor methodological errors and text editing shown in the manuscript.

Reviewer 3 Report

The work is of interest. A large set of research methods is presented in the manuscript. These studies will undoubtedly be of interest to a wide readership. The formation of the composition and structure has been considered, and the mechanical, tribological properties, corrosion resistance, density and surface roughness of Ti-Zr-Mo alloys have been studied.

A large volume of the data has been obtained allowing estimating Ti-Zr-Mo alloys. The main dimensional characteristics for the grains have been presented: an average chemical composition of the β solid solution made from as-deposited Ti-Zr-Mo alloys, the average size of β grains in as-deposited Ti-Zr-Mo alloys, etc. A fairly large number of literature sources have been provided.

However, there are some questions and recommendations:

(1) The authors are recommended to get acquainted with the work “Diffusivities and Atomic Mobilities in bcc Ti-Mo-Zr Alloys Materials” 2018, 11, 1909; doi:10.3390/ma11101909. At the discretion of the authors.

(2) The Abstract cites “TA2 substrates” and it is not entirely clear what they are composed of. Please, provide information that is more precise.

(3) Please, interpret the abbreviations: line 194 “EBSD”.

(4) Figure 5e below shows that the surface is slightly different. Have you conducted studies on the composition and structure in this area?

(5) Line 240 has “and i is”. What is “i”?

6) The authors laid down the most important conclusions based on the results of the work. The conclusions are clear. Perhaps readers will be more interested if you add a little more detail in the Conclusions.

Author Response

Dear Reviewer,

Thank you for your useful comments and suggestions. We have modified the manuscript accordingly, and detailed corrections are listed below:

1. The authors are recommended to get acquainted with the work “Diffusivities and Atomic Mobilities in bcc Ti-Mo-Zr Alloys” Materials, 2018, 11, 1909; doi:10.3390 /ma11101909. At the discretion of the authors.

Thanks for the recommendation, the work has been cited.

2. The Abstract cites “TA2 substrates” and it is not entirely clear what they are composed of. Please, provide information that is more precise.

The TA2 was misused. Actually, the substrate material used for L-DED is high purity titanium with mass purity of 99.90% rather than commercial pure titanium TA2. This error has been corrected in revised manuscript.  

3. Please, interpret the abbreviations: line 194 “EBSD”.

The abbreviation “EBSD” has been appended to its full name in experimental procedures section. 

4. Figure 5e below shows that the surface is slightly different. Have you conducted studies on the composition and structure in this area?

The relevant content has been added to microstructure section.

5. Line 240 has “and i is”. What is “i”?

i refers to the individual solute in the multicomponent system, which has been explained in revised manuscript.

6. The authors laid down the most important conclusions based on the results of the work. The conclusions are clear. Perhaps readers will be more interested if you add a little more detail in the Conclusions.

According to the comment, more details have been added to the conclusions.

Round 2

Reviewer 1 Report

Accept in present form